# Bending Performance of Steel Fiber Reinforced Concrete Beams Based on Composite-Recycled Aggregate and Matched with 500 MPa Rebars

**DOI:** 10.3390/ma13040930

**Published:** 2020-02-19

**Authors:** Xiaoke Li, Songwei Pei, Kunpeng Fan, Haibin Geng, Fenglan Li

**Affiliations:** 1School of Civil Engineering and Communications, North China University of Water Resources and Electric Power, Zhengzhou 450045, China; psw@ncwu.edu.cn; 2International Joint Research Lab for Eco-building Materials and Engineering of Henan, North China University of Water Resources and Electric Power, Zhengzhou 450045, China; kunpeng_mail@163.com (K.F.); hbgeng@stu.ncwu.edu.cn (H.G.)

**Keywords:** steel fiber reinforced composite-recycled aggregate concrete (SFR-CRAC), beam, high-strength rebar, cracking moment, crack width, flexural stiffness, flexural capacity, flexural ductility

## Abstract

To promote the engineering application of recycled aggregate for concrete production with good adaptability and economic efficiency, this paper performed a campaign to investigate the flexural performance of steel fiber reinforced composite-recycled aggregate concrete (SFR-CRAC) beams matched with 500 MPa longitudinal rebars. The composite-recycled aggregate has features of the full use recycled fine aggregate and small particle recycled coarse aggregate, and the continuous grading of coarse aggregate ensured by admixing the large particle natural aggregate about 35% to 45% in mass of total coarse aggregate. The properties of SFR-CRAC have been comprehensively improved by using steel fibers. With a varying volume fraction of steel fiber from 0% to 2.0%, 10 beam specimens were produced. The flexural behaviors of the beams during the complete loading procedure were experimentally studied under a four-point bending test. Of which the concrete strain at mid-span section, the appearance of cracks, the crack distribution and crack width, the mid-span deflection, the tensile strain of longitudinal rebars, and the failure patterns of the beams were measured in detail. Results indicated that the assumption of plane cross-section held true approximately, the 500 MPa longitudinal rebars worked at a high stress level within the limit width of cracks on reinforced SFR-CRAC beams at the normal serviceability, and the typical failure occurred with the yield of 500 MPa longitudinal rebars followed by the crushed SFR-CRAC in compression. The cracking resistance, the flexural capacity, and the flexural ductility of the beams increased with the volume fraction of steel fiber, while the crack width and mid-span deflection obviously decreased. Finally, by linking to those for conventional reinforced concrete beams, formulas are suggested for predicting the cracking moment, crack width, and flexural stiffness at normal serviceability, and the ultimate moment at bearing capacity of reinforced SFR-CRAC beams.

## 1. Introduction

Coming into 21th century, a lot of research has been attracted to solving the issue that construction waste surrounds cities in the process of large-scale urban conversion and renewal, of which the reuse of recycled aggregates made of the waste of mining extraction and the demolished old concretes became a hot point in the resource utilization of construction waste. This has spawned a new branch of eco-building materials and the renovation of concrete production equipment dealing with the recycled aggregate. Many recycling and treatment technologies are investigated to improve the morphology feature and surface defects of recycled coarse aggregate [1,2,3,4,5], the design methods for the mix proportion of concrete is modified to concern about the changes of performance due to the natural aggregate partially or fully substituted by recycled aggregate [5,6,7,8,9], and the mixing method of mixture is perfected with pre-soaked procedure to face the issue of high water absorption of recycled aggregates [5,10,11,12,13]. Generally, these researches provide a foundation for the engineering application of concrete with recycled aggregate.

Referred to the experiences of the above researches, an innovative idea of the composite-recycled aggregate (CRA) was formed, and scientific research was carried out step by step to let the engineering concrete production with CRA has good adaptability and economic efficiency. The feature of the CRA is the full use of recycled fine and coarse aggregates, of which the particle size of recycled coarse aggregate is made smaller than that of its mother aggregate in demolished old concrete, and a certain amount of large particle natural aggregate is admixed with the requirement of the continuous grading of coarse aggregate. This avoids the secondary pollution caused by waste recycled fine aggregate, and solves the defects of concrete with large particle recycled coarse aggregate. With proper mix proportion and rational process, the concrete with CRA, abbreviated as CRAC, has been successfully produced with good workability and expected mechanical properties [13,14,15,16,17]. Combined with the technology of high-performance concrete reinforced with steel fibers, steel fiber reinforced composite-recycled aggregate concrete (SFR-CRAC) was also developed to comprehensively promote the performance of CRAC [18,19,20]. Meanwhile, the research on the structural application of SFR-CRAC was also conducted. The experimental investigation of reinforced SFR-CRAC columns under large eccentric compression indicated that [21,22], the cracking resistance of the columns increased due to the increase of tensile strength of SFR-CRAC by the presence of steel fibers, the crack spacing decreased with the uniform distribution of the SFR-CRAC strains, and the crack width decreased with the lower tensile stress of the longitudinal steel bars; a higher bearing capability after the ultimate load leads to a better ductility of the columns with large lateral deformation, and the ductility increases with an increase of the volume fraction of steel fiber. Based on the principle linking those of conventional reinforced concrete columns, the predictive formulas of crack width and bearing capacity of reinforced SFR-CRAC columns were proposed.

In view of the engineering structural application, the bending performances of reinforced SFR-CRAC flexural members such as beams and plates also need to be studied [23]. This is, in fact, a basic work for the new kinds of concrete applied in structures; for example, the recycled aggregate concrete, the steel fiber reinforced expanded-shale lightweight concrete [24,25,26]. Therefore, to further verify the applicability and adaptability of SFR-CRAC for the concrete structures, the loading performances of reinforced SFR-CRAC flexural members should be confirmed with the comparison of current design methods specified in codes [23,27,28,29].

## 2. Research Significance

Based on the above statements, it is necessary to understand the bending performance of reinforced SFR-CRAC flexural members. Meanwhile, as the main grade of steel bar used for conventional reinforced concrete structures, the performance of 500MPa hot-rolled deformed rebar matched with SFR-CRAC also needs to be certified [23]. Therefore, this paper performed a campaign to study the bending behavior of reinforced SFR-CRAC beams, of which the volume fraction of steel fiber is selected as the main parameter. Eight reinforced SFR-CRAC beams referenced by two reinforced CRAC beams were designed and tested under the four-point bending test. The flexural performance of the test beams while the complete loading procedure was measured. The cracking resistance, crack distribution pattern and crack width, mid-span deflection and flexural ductility, and failure pattern and bearing capacity of the test beams are analyzed on the basis of the experiment. The adaptability of 500 MPa rebar used as longitudinal tensile reinforcement of SFR-CRAC beams is discussed in view of the limit crack width at normal serviceability and the ultimate moment at bearing capacity. Finally, the design method for reinforced SFR-CRAC beams is proposed in accordance with current design codes for reinforced concrete beams. 

## 3. Experimental Work

### 3.1. Design of the Test Beams

Based on the principle that test beams failed with the yield of longitudinal tensile reinforcement followed by the crushed concrete in the compression zone [23,29], ten test beams with a rectangular cross-section were designed (width *b* = 150 mm, depth *h* = 300 mm, and length *l* = 2.7 m). The span was *l*_0_ = 2.4 m with a pure bending length of 1.0 m. Details of the test beams are presented in Figure 1. Two of them were the same as a group. Compared with the composite-recycled aggregate concrete (CRAC) in a strength grade of C40, the steel fiber reinforced composite-recycled aggregate concrete (SFR-CRAC) was designed in a strength grade of FC40 with a varying volume fraction of steel fiber *v*_f_ = 0.8%, 1.2%, 1.6%, and 2.0%, respectively. The longitudinal tensile reinforcement was an HRB500 hot-rolled deformed steel bar with diameter *d* = 16 mm, the tested yield strength *f*_y_ = 560 MPa, the ultimate tensile strength *f*_st_ = 707 MPa, and the modulus of elasticity *E*_s_ = 2.05 × 10^5^ MPa. The longitudinal construction reinforcement was an HPB300 hot-rolled plain steel bar with a diameter of 12 mm. The stirrups were an HPB300 hot-rolled plain steel bar with a diameter of 8 mm; the tested yield strength and the ultimate tensile strength were 425 MPa and 500 MPa. The thickness of concrete cover for longitudinal tensile rebars was *c*_s_ = 25 mm, and the ratio of longitudinal tensile rebars was *ρ* = 1.03%.

### 3.2. Preparation of SFR-CRAC

For the preparation of SFR-CRAC, ordinary silicate cement in a strength grade of 42.5, which was produced by Zhengzhou Tianrui Cement Co. Ltd., China, was used as the binder [16]. The additive was the high-performance polycarboxylic acid water reducer, which produced by Jiangsu Sobute New Materials Co. Ltd., China. The mix water was tap water. Steel fiber was of mill-cut type in length *l*_f_ = 32 mm and equivalent diameter *d*_f_ = 0.8 mm, and the tensile strength was over 600 MPa, which was produced by Shanghai Harax Steel Fiber Technology Co. Ltd., China. 

Due to the great impaction of the morphology of coarse aggregate on the fresh workability and the hardened performances of concrete, the production of recycled aggregate went through a three-stage process [1,3,9,30]. The primary and secondary crushing of demolished concrete blocks and crushed particles were disposed during the first and second stages by a jaw crusher, and the rational particle shape and grading of the recycled aggregate product was performed in the third stage by a vertical impact crusher. The crushers are produced by Zhengzhou Dingsheng Engineering Technology Co. Ltd. China. The coarse aggregate was composited by the recycled coarse aggregate with a particle size of 5–16 mm and the natural aggregate with a particle size of 16–25 mm. To meet the requirement of continuous particle distribution in accordance with the specifications of China codes [31,32], the amount of natural aggregate was 40% in mass of total coarse aggregate based on the optimization of the previous study [16,21,22]. The fine aggregate was the recycled fine aggregate which was the byproduct of recycled coarse aggregate; the gradation with a particle size less than 5 mm met the requirement specified in China code GB/T25176 [33]. The basic properties of aggregates are presented in Table 1, of which the identifier RA is recycled aggregate, and NA is natural aggregate. The water absorption of recycled aggregates at 10 min was over 90% for 24 h.

The mix proportion of SFR-CRAC and referenced CRAC was designed by the absolute volume method on the condition of surface-drying saturated aggregates as per China codes JG/T 472 and JGJ55 [34,35]. The water-to-cement ratio *w/c* = 0.49 with a kept cement dosage of 409 kg/m^3^, and the recycled fine aggregate was 42% in mass of total aggregates. For the mix proportion of SFR-CRAC, the mass of steel fibers was counted into the total mass of aggregates, and the dosage of additive was increased with the volume fraction of steel fiber to ensure the rational workability of fresh SFR-CRAC. The additional water was added appropriately based on the tested water absorption of recycled aggregates. After adjustment, the mix proportions were determined as presented in Table 2. 

### 3.3. Fabrication of the Test Beams

The test beams were produced with the following procedure: (1) preparing the raw materials of SFR-CRAC and rebars, (2) fabricating the rebars to be a skeleton, (3) placing the rebar skeleton into the steel mold, (4) casting and compacting the mixture of SFR-CRAC in the steel mold, and (5) demolding and curing.

The single horizontal shaft forced mixer was used to mix the fresh mixture. The aggregates were firstly pre-soaked with the additional water in the mixer for 1 h, then the cement and half dosage of mix water were added and mixed for 30 s. During the mixing, water reducer and another half dosage of mix water were added. After that, the steel fiber was sprinkled into the mixer and mixed for 3 min.

The fresh mixture was cast into the steel mold of the test beams and compacted with vibrators which fixed to the out-sides of the steel mold. After casting and until demolding, the screed top surface was covered by a plastic film for 48 h. After demolding, the test beams were cured with sprayed water for 7 d, then placed in natural condition at least 19 d before testing. As per China standards JG/T472 and GB50152 [30,32], standard specimens of six cubes with dimensions of 150 mm and six prisms with dimensions of 150 mm × 150 mm × 300 mm were made at the same time and cured in the same condition accompanying with each group of the test beams. Three of the cubes as a group were tested for the cubic compressive strength *f*_cu_ and the splitting tensile strengths *f*_ft_ of SFR-CRAC. Three of the prisms as a group were tested for the axial compressive strength *f*_fc_ and the modulus of elasticity *E*_c_ of SFR-CRAC. Results are presented in Table 2. 

### 3.4. Test Methods

The four-point bending test method was used in this study according to the specification of China standard GB50152 [36]. Two symmetrical concentrated loads were exerted on the top surface of the test beams. The loading device consisting of the steel frame, hydraulic jack, and distributive girder, and the loads were controlled by the load sensors. The mid-span deflection was measured by the electrical displacement meters installed at the mid-span, loading sections, and supports [26,37]. To verify the assumption of plane cross-section, six electrical strain meters were arranged along the depth of the mid-span cross-section. Cracking load was determined through the conditions that the first crack(s) appeared on the side surface of the test beams at the barycenter of longitudinal tensile rebars, and the first point changed slope at the load-deflection curve. Crack width on side surfaces of the test beams at the barycenter of longitudinal tensile rebar was measured by the electrical reading microscope with 0.02 mm precision. The strain of longitudinal tensile rebars was measured by the strain gauges with a length of 1 mm and an electric resistance of 120 Ω. The strain gauges were pasted on each of the longitudinal tensile rebars in space of 268 mm within the pure bending segment, as presented in Figure 2. All the data were collected by a data acquisition system in the lab.

## 4. Test Results

### 4.1. Strain at Mid-Span Cross-Section

Figure 3 presents the concrete strain along the depth of the mid-span section at different bending moment *M*. With the increase of loads, the sectional neutral axis moved up to reduce the depth of the compressive zone and enlarge the tensile zone of the cross-section. This trend was more obvious on beams with a higher volume fraction of steel fiber such as the test beams RFB-2.0a/b. The concrete strain varied approximately to be linear along the cross-section from top to bottom, which basically satisfied the assumption of plane cross-section [25,27,28,29,30]. This provides the basis of theoretical prediction of the cracking resistance, the tensile stress of longitudinal rebar, and the bearing capacity of the test beams in accordance with the principles of Materials Mechanics. 

### 4.2. Tensile Strain of Longitudinal Rebars

The average tensile strains of longitudinal rebars at or near the cracked sections within the pure bending segment of each test beam are exhibited in Figure 4. Three parts of the curves are obviously exhibited with the increase of bending moment *M*. The first linear part started from initial loading to the cracking of concrete. In this part, the rebars have a function to enlarge the geometrical peculiarity of the concrete section to strengthen the cracking resistance of reinforced concrete beams [38]. After that, the tensile strain of the rebar grew fast into the second part until the point of the yield of rebars. The feature point referred to the yield strain *ε*_s_ = *f*_y_/*E*_s_= 2732με, and different variations appeared between the reinforced SFR-CRAC beams and the reinforced CRAC beams. The strain growth on the reinforced SFR-CRAC beams became slower with the aids of steel fibers bridging cracks, and large nonlinearity appeared with the increase of the *v*_f_. This indicated that steel fiber has a function bearing tensile stress in the sectional tensile zone. 

The third part turned up due to the yield of the rebars; the strain quickly grew with small increased loads. This corresponds to the stress–strain relationship of the rebar after yielding. With proper reinforcement ratio, the yield strength of the rebar can be used to compute the bearing capacity of the test beams.

Based on the design principle of the ultimate states for reinforced concrete beams, the crack width and mid-span deflection at normal serviceability should be checked. This corresponds to the loading level *M*/*M*_u_ = 0.45–0.75 considering different combinations of dead loads and live loads on the structures. Here *M*_u_ is the ultimate moment at bearing capacity of the cross-section [23,38]. As presented in Figure 5, the tensile strain of the longitudinal rebar increases almost linearly to the loading level in this range. At the same loading level *M*/*M*_u_, the tensile strain trends to decrease with the increasing *v*_f_. Therefore, by using the tensile strength of SFR-CRAC and considering the direct proportionality of stress with strain of the rebar, the predictive formula of the rebar tensile stress *σ*_s_ can be built by modifying that of reinforced concrete beams [23,39], that is:(1)$$\sigma_{s}=M/(0.87h_{0}A_{s})-2f_{\mathrm{ft}}ba_{s}/A_{s}$$
where *M* is the moment at normal service stage; *a*_s_ is the distance from the bottom surface of beam to the barycenter of longitudinal tensile rebars, *a*_s_ = *c*_s_ + *d*/2 in this test; *f*_ft_ is the tensile strength of SFR-CRAC. *f*_ft_ = 0 for the CRAC.

Table 3 presents the test results of the rebar tensile stress *σ*_s_ at loading level *M*/*M*_u_ = 0.45–0.75. The ratios between tested and calculated values of the *σ*_s_ are exhibited in Figure 6. For the tested reinforced SFR-CRAC beams and the referenced reinforced CRAC beams, the average of the ratios is 0.958 and 1.005, with a dispersion coefficient of 0.098 and 0.103, respectively. 

### 4.3. Crack Distribution and Failure Pattern

As exhibited in Figure 7, typical crack distribution appeared within the pure bending segment, and the failure turned up with the yield of longitudinal tensile rebars followed by the crushed SFR-CRAC in the compressive zone, accompanied by the rapid widening of main cracks. Table 3 presents the main tested data of the cracking moment *M*_cr_, the ultimate moment *M*_u_, and the average crack width *w*_m_, maximum crack width *w_max_* and mid-span deflection *a*_f_ at normal service stage corresponded to the loading level *M*/*M*_u_ = 0.45−0.75. 

The initial crack appeared at a random cross-section, which controlled the cracking resistance of the test beams. Based on test data in Table 3, the moment ratio of cracking to ultimate *M*_cr_/*M*_u_ of the test beams increased from 18% to 22% with the increase of the *v*_f_ from 0% to 2.0%. This indicated a higher enhancement on cracking resistance than bearing capacity by the presence of steel fiber.

New cracks turned up between adjacent cracks until the tensile strain was no longer over the ultimate of SFR-CRAC. After that, the crack spacing became steady, and the main cracks grew in width with the increased bending moment. Due to the random appearance and the different widening of cracks, the variation of crack distribution of reinforced concrete beams at normal serviceability can be described by the average crack spacing and the enlarge coefficient of maximum width from average width [38,39]. Based on test data presented in Table 4, due to more cracks appearing on the pure bending segment, the crack spacing of reinforced SFR-CRAC beams tends to decrease with the increasing *v*_f_. By using the crack spacing of conventional reinforced concrete beams as the basis to evaluate the effectiveness of steel fibers, the average crack spacing *l*_fcr_ of reinforced SFR-CRAC beams can be calculated as:(2)$$l_{\mathrm{fcr}}=(1-0.1\lambda_{f})(1.9c_{s}+0.08d/\rho_{\mathrm{te}})$$
(3)$$\rho_{\mathrm{te}}=A_{s}/\left(0.5bh\right)$$
where *ρ*_te_ is the effective reinforcement ratio of longitudinal tensile rebars; *λ*_f_ is the fiber factor, the product of *v*_f_ with *l*_f_/*d*_f_ of steel fiber.

With the comparison as presented in Table 4, a good agreement is given out between the tested and the calculated values of crack spacing by the average ratio of 1.046, with a dispersion coefficient of 0.036.

The dispersion of crack width reflects the transferability of tensile stress across cracks, which can be statistically analyzed by the ratios between the width of each crack and the average width of all cracks (*w*_i_/*w*_m_) on a test beam [39,40]. Considering the main cracks on the reinforced SFR-CRAC beams, the frequency of the ratios is counted as exhibited in Figure 8. Sixty-one main cracks appeared on the test beams with 186 test data of crack width. After analysis, the frequency histogram of *w*_i_/*w*_m_ fits the normal distribution. Taking the assurance rate at 95%, the enlarging coefficient of maximum width from average width *τ*_s_ = 0.991 + 1.645 × 0.376 = 1.61. This is slightly small compared with that *τ*_s_ = 1.66 for the conventional reinforced concrete beams [38]. The reduction of the enlarging coefficient means a uniform distribution of crack width on the reinforced SFR-CRAC beams. 

It can be seen from Table 3 that the maximum crack width was up to 0.34 mm for reinforced CRAC beams, while the tensile stress of longitudinal rebar was 46.4–83.8% of yield strength. The presence of steel fibers could reduce the crack width lower than 0.25 mm with the tensile stress of longitudinal rebars at 34.4–77.9% yield strength. This indicates the reinforced SFR-CRAC beams can satisfy a stricter limit of crack width to meet the durability of concrete structures [41]. Meanwhile, due to the increase of the ultimate moment at bearing capacity, the reinforced SFR-CRAC beams have higher loading capacity at normal serviceability. For example, compared with beams RFB-0a/b, beams RFB-2.0a/b had a 16.4% increment of the ultimate moment, with the equal increment the moment at normal serviceability can be increased by 16.4%. This benefits to the improvement of the normal loading capacity of reinforced SFR-CRAC beams on conditions satisfying both of normal serviceability and ultimate bearing capacity.

### 4.4. Mid-span Deflection and Flexural Ductility

As presented in Figure 9, four segments can be seen from the curves of mid-span deflection changing with the moment. The first is linear with elastic deformation of SFR-CRAC before cracking. The second is almost linear with the crack development; this belongs to the normal serviceability of the test beams. The third turns up with obvious nonlinearity due to the plastic compression and the rapid development of cracks followed with the yield of longitudinal tensile rebars. This presents to the ultimate state of bearing capacity of the test beams. The fourth exhibits the residual loading capacity with large plastic deformation after yield of longitudinal tensile rebars; the gentle slope and greater deflection reflect the beneficial effect of steel fiber on the flexural ductility, and the mid-span deflection of the test beams at failure state increases with the *v*_f_.

The flexural ductility of reinforced concrete beams is always defined as the ratio of deflection *δ*_i_ at a post-peak moment to that *δ*_y_ corresponding to the yield of longitudinal tensile rebar. Of which, *δ*_0.85_, *δ*_0.95_, and *δ*_1.0_ were normally taken as the deflection at the post-peak moment of 0.85*M*_u_, 0.95*M*_u_, and *M*_u_ [26,42]. The flexural ductility index *β*_i_ is computed as follows:(4)$$\beta_{i}=\delta_{i}/\delta_{y}\;(i=0.85,\;0.95,\;1.0)$$

The results are presented in Table 5 and Figure 10. In spite of the dispersion of test data, nonlinear increasing of *β*_i_ exists with the increase of *v*_f_ no matter what level of the post-peak moment is adopted. The slight decrease of *β*_1.0_ indicates the increase of flexural stiffness at peak moment due to the enhancement of steel fiber bridging the cracks in the tensile zone. Compared to the reinforced CRAC beams, the flexural ductility indexes *β*_0.85_ and *β*_0.95_ of reinforced SFR-CRAC beams with *v*_f_ = 2.0% increased by about 204% and 154% on average. The promotion of *β*_0.85_ over *β*_0.95_ implies the flexural ductility at the post-peak moment was improved significantly by the presence of steel fibers with larger *v*_f_. Moreover, the increase of flexural ductility index *β*_0.85_ with *v*_f_ demonstrates the promotion of the post-peak loading sustainability of reinforced SFR-CRAC beams due to the benefits of steel fibers bridging the cracks and keeping the entirety of compression zone. 

## 5. Discussion

### 5.1. Cracking Resistance

The cracking resistance of reinforced SFR-CRAC trends to increase with the *v*_f_. As presented in Figure 11, almost the same increments of cracking moment *M*_fcr_ and tensile strength *f*_ft_ are produced with similar growth rates of 0.19 and 0.18. This indicates the effect of steel fiber on the cracking resistance of reinforced SFR-CRAC beams depends on the enhancement of tensile strength of SFR-CRAC, and the increment of *M*_fcr_ of reinforced SFR-CRAC beams can be predicted by the increment of *f*_ft_ with the same *v*_f_.

Based on the principle of Materials Mechanics, and considering the beneficial effect of subsistent plasticity of SFR-CRAC in the tensile zone of the cross-section, the *M*_fcr_ of reinforced SFR-CRAC beam can be predicted by the following formulas [26,38,39]: (5)$$M_{\mathrm{fcr}}=\gamma f_{\mathrm{ft}}W_{0}$$
(6)$$\gamma =1.55(0.73+\frac{60}{h})$$
where *γ* is the sectional plasticity coefficient of resistance-moment; *W*_0_ is the elastic resistance-moment to the tensile edge of transformed section.
(7)$$W_{0}=I_{0}/(h-y_{0})$$
(8)$$y_{0}=\frac{\frac{bh^{2}}{2}+\alpha_{E}A_{s}h_{0}}{bh+\alpha_{E}A_{s}}$$
(9)$$I_{0}=\frac{by_{0}^{3}}{3}+\frac{b{\left(h-y_{0}\right)}^{3}}{3}+\alpha_{E}A_{s}{\left(h_{0}-y_{0}\right)}^{2}$$
where *I*_0_ is the inertia moment of transformed section to its centroid; *y*_0_ is the distance of centroid to compressive edge of transformed section; *α*_E_ is the ratio of the modulus of elasticity of the rebar to that of SFR-CRAC; *A*_s_ is the sectional area of longitudinal tensile rebars.

By using Formula (5), the ratio of test value to the calculated value of *M*_fcr_ changes from 0.992 to 1.132, and the mean ratio is 1.046, with a dispersion coefficient of 0.036. A good agreement between tested and calculated values are presented in Figure 12.

### 5.2. Crack Width

Combined with the bond-slip model of cracks on the reinforced concrete beams, steel fibers in the tensile zone improve the transferability of tensile stress of concrete across cracks, and assist the tensile rebars to confine the opening of cracks. Therefore, except for the improvement of crack distribution by shortening the crack spacing *l*_fcr_ and reducing the enlarging coefficient *τ*_s_ of maximum crack width *w*_max_ from average crack width *w*_m_, steel fibers also adjust the uneven strain distributions of SFR-CRAC and longitudinal tensile rebar among cracks. The coefficient *α*_c_ reflecting the uneven deformation of SFR-CRAC between cracks becomes bigger, and the coefficient *ψ* reflecting the uneven strain distribution of longitudinal tensile rebar among cracks becomes smaller. The predictive formulas for the maximum and average crack width on the side surface of reinforced SFR-CRAC beams at the barycenter of longitudinal tensile rebars can be written as follows [37,38,39]: (10)$$w_{\max}=\tau_{s}w_{m}$$
(11)$$w_{m}=\alpha_{c}\psi \frac{\sigma_{s}}{E_{s}}l_{\mathrm{fcr}}$$
(12)$$\psi =1.1-\frac{0.65f_{\mathrm{ft}}}{\rho_{\mathrm{te}}\sigma_{s}}$$

Based on test data of this study, *α*_c_ = 0.85 for referenced reinforced CRAC beams, and *α*_c_ = 0.95 for reinforced SFR-CRAC beams. The larger *α*_c_ means the smaller uneven deformation of SFR-CRAC between cracks. The ratios between tested values in Table 3 and calculated values with Formula (11) of the average crack width, and Formula (10) of the maximum crack width are exhibited in Figure 13. For the referenced reinforced CRAC beams, the averaged ratios of the average crack width and the maximum crack width are 1.045 and 1.059, with a dispersion coefficient of 0.180 and 0.111, respectively. For the reinforced SFR-CRAC beams, the averaged ratios of the average crack width and the maximum crack width are 1.027 and 0.967, with a dispersion coefficient of 0.173 and 0.176, respectively. 

### 5.3. Flexural Stiffness

Based on the principle of Materials Mechanics, the mid-span deflections of the reinforced SFR0CRAC beams can be computed as:(13)$$a_{f}=0.1065{Ml_{0}}^{2}/B_{s}$$
where *B*_s_ is the equivalent flexural stiffness of beam.

Figure 14 presents the tested *B*_s_^t^ computed with *a*_f_^t^ in Table 3. Corresponding to the changes of *a*_f_^t^, *B*_s_^t^ decreased with the increase of load level *M*/*M*_u_, and had an uptrend with *v*_f_. 

Referring to that of steel fiber reinforced expanded-shale lightweight concrete beams [26], the equivalent flexural stiffness of reinforced SFR-CRAC beams at mid-span can be calculated as: (14)$$B_{s}=\frac{\alpha_{B}E_{c}I_{0}}{1+(1.16-M_{\mathrm{cr}}/M)/(6\alpha_{E}\rho )}$$
where *α*_B_ is the coefficient reflecting the initial damages on the elastic flexural stiffness.

Based on test data of this study, *α*_B_ = 0.85 for referenced reinforced CRAC beams, and *α*_B_ = 0.95 for reinforced SFR-CRAC beams. This indicates the smaller initial damages of reinforced SFR-CRAC beams. The ratios of tested to calculated flexural stiffness *B*_s_^t^/*B*_s_ are displayed in Figure 15. For the referenced reinforced CRAC beams, the ratios change from 1.030 to 1.294, and the mean value is 1.112, with a dispersion coefficient of 0.084. For reinforced SFR-CRAC beams, the ratios change from 0.924 to 1.553, and the mean value is 1.085, with a dispersion coefficient of 0.144. 

### 5.4. Flexural Capacity

Considering the beneficial effect of steel fibers in tensile zone of cross-section, and on the promise of keeping the numerical diagram of moment capacity for conventional reinforced concrete beams, as seen in Figure 16, the strengthening effect of steel fibers in tensile zone is simplified as an equivalent rectangular with the effective stress *f*_ftu_ at a depth of *x*_t_ [25,26,27]. Based on the principle of force and moment equilibrium at the cross-section, the ultimate moment *M*_fu_ of reinforced SFR-CRAC beams can be calculated with the formulas as follows:(15)$$\alpha_{1}f_{\mathrm{fc}}bx=f_{y}A_{s}+f_{\mathrm{ftu}}bx_{t}$$
(16)$$M_{\mathrm{fu}}=f_{y}A_{s}(h_{0}-\frac{x}{2})+f_{\mathrm{ftu}}bx_{t}(h-\frac{x}{2}-\frac{x_{t}}{2})$$
(17)$$x_{t}=h-\frac{x}{\beta_{1}}$$
(18)$$f_{\mathrm{ftu}}=\beta_{\mathrm{tu}}\lambda_{f}f_{t}$$
where *f*_t_ is the tensile strength of CRAC with the same strength grade of SFR-CRAC; *x* is the compression depth of equivalent rectangular stress block; *α*_1_ is a coefficient of equivalent rectangular stress; *β*_1_ is a coefficient about the depth of equivalent rectangular. In this study, *α*_1_ = 1.0 and *β*_1_ = 0.75 for the strength *f*_cu_ < 40 MPa [23,38].

Combined with the test data, and resolve the formulas to get the *x*, and then *β*_tu_ could be got. Take the mean value of *β*_tu_ = 1.15. The comparison of tested and calculated values of *M*_fu_ is presented in Figure 17. A good agreement is given out with the mean ratio of tested to calculated values of 0.986, with a dispersion coefficient of 0.034.

The test data is also verified by the calculation model of ACI544.4R considering the enhancement of steel fiber in the tensile zone of the beams [28]. As presented in Figure 17, a mean ratio of tested to calculated values is 1.104, with a dispersion coefficient of 0.061. It shows that the calculated moment was obviously lower than the test value. 

## 6. Conclusions

Based on the study of this paper, the conclusions can be drawn as follows: (1)Steel fiber reinforced composite-recycled aggregate concrete (SFR-CRAC) prepared in this study has good workability to produce the reinforced SFR-CRAC beams.(2)Similar to conventional reinforced concrete beams, the assumption of plane cross-section is valid to build the calculation models of cracking resistance, crack width, flexural stiffness, and flexural capacity of the reinforced SFR-CRAC beams.(3)The same enhancement of steel fibers on the cracking resistance of reinforced SFR-CRAC beams and the tensile strength of SFR-CRAC. The cracking moment of reinforced SFR-CRAC beams can be predicted by using the tensile strength of SFR-CRAC. The reduced crack spacing and good crack distribution pattern of reinforced SFR-CRAC benefit to minimize the crack width. This also improves the strain distribution pattern and reduces the tensile stress of longitudinal tensile rebars at cracked sections. Therefore, the 500 MPa longitudinal tensile rebars work at a high stress level with the premise of matching the limit crack width specified in the standard for reinforced SFR-CRAC beams at normal serviceability state. The flexural stiffness of reinforced SFR-CRAC beams at normal serviceability also increased with the improvement of crack distribution.(4)Designed with the rational longitudinal reinforcement ratio, the reinforced SFR-CRAC beams fail in the typical mode of the yield of 500 MPa longitudinal rebars followed by the fast widening of main cracks and the crushed SFR-CRAC in the compression zone. Good flexural ductility presents on the reinforced SFR-CRAC beams matched with 500 MPa longitudinal rebars, especially at the post-peak loading levels.(5)Considering the improvements by steel fibers and linked with those of conventional reinforced concrete beams, formulas for predicting the cracking moment, crack width, flexural stiffness, and ultimate moment of SFR-CRAC beams are proposed for the reference of design.

## Figures and Tables

**Figure 1 materials-13-00930-f001:**
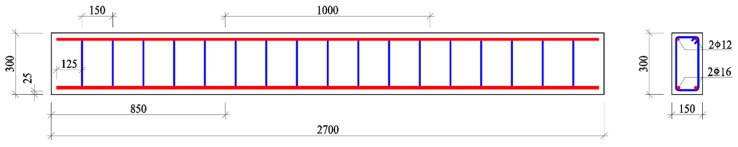
Details of the test beams.

**Figure 2 materials-13-00930-f002:**
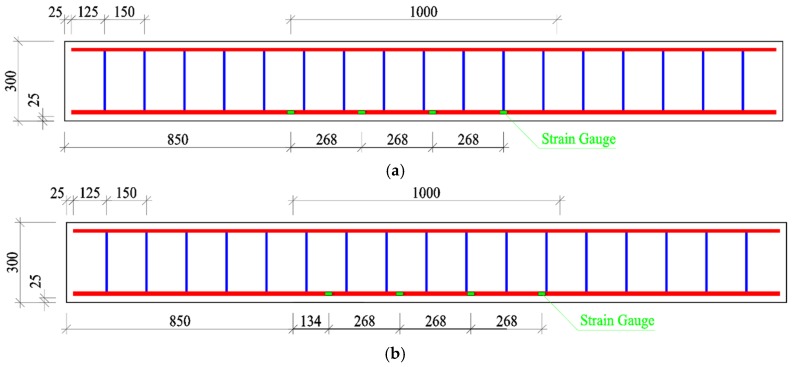
Arrangement of strain gauges on: (**a**) one of the longitudinal tensile rebars; (**b**) another longitudinal tensile rebar.

**Figure 3 materials-13-00930-f003:**
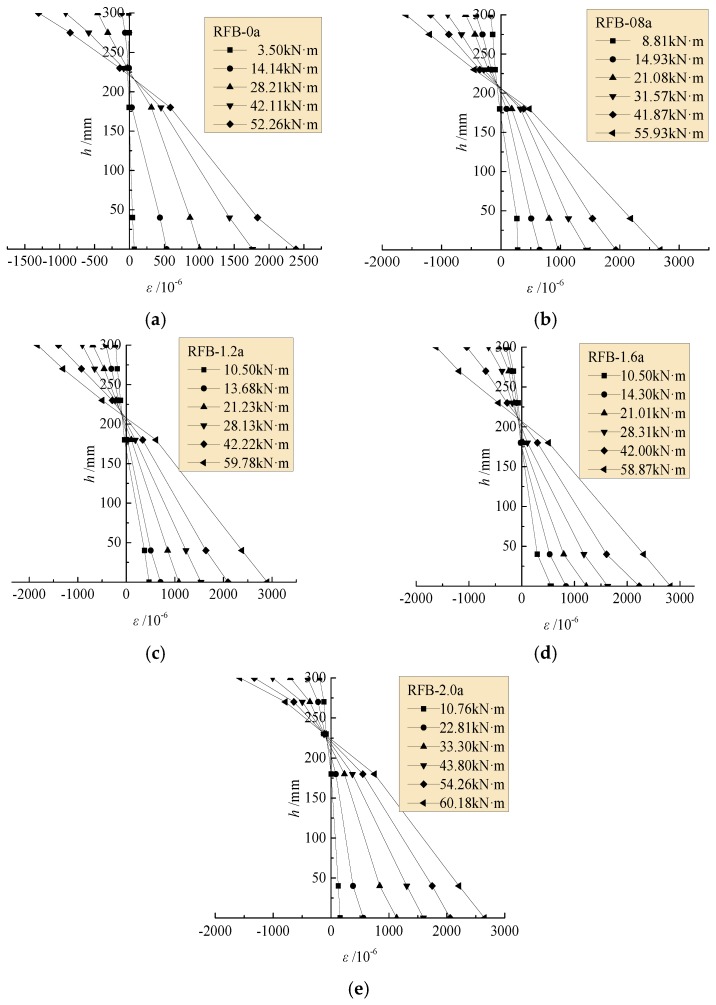
Concrete strain along depth of the mid-span section: (**a**) RFB-0a; (**b**) RFB-0.8a; (**c**) RFB-1.2a; (**d**) RFB-1.6a; (**e**) RFB-2.0a.

**Figure 4 materials-13-00930-f004:**
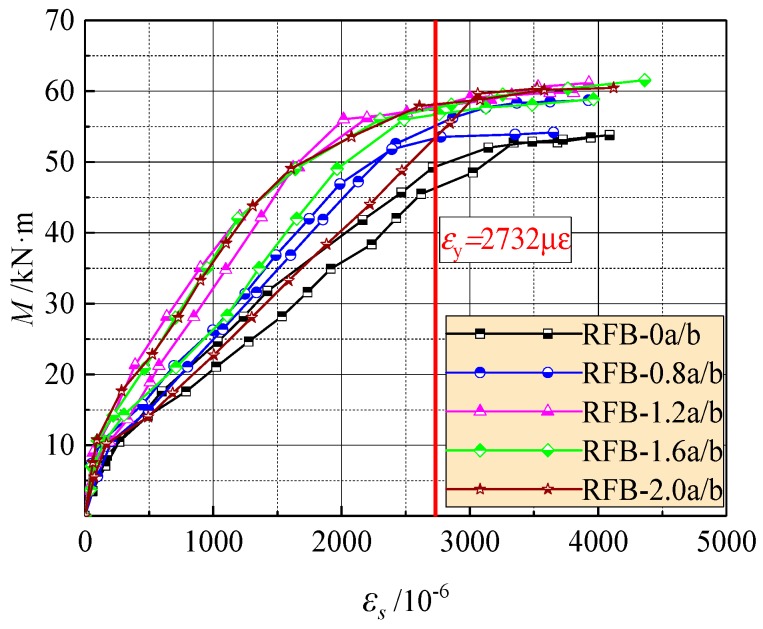
Changes of the strain of longitudinal tensile rebars with moment on the pure bending sections.

**Figure 5 materials-13-00930-f005:**
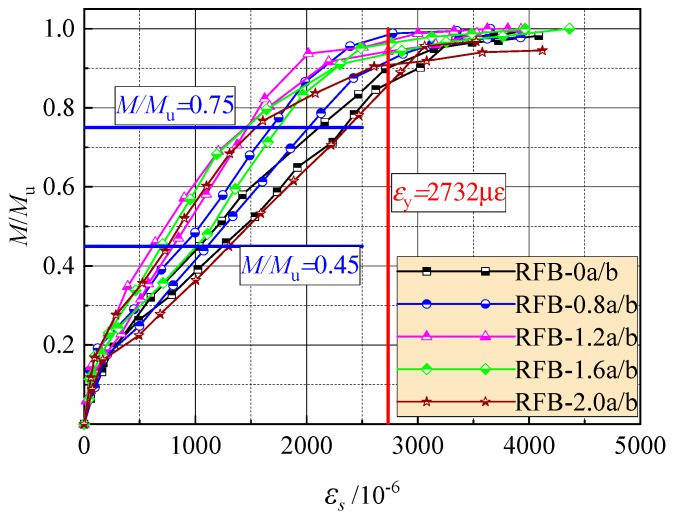
Strain of the longitudinal tensile rebar at different loading levels.

**Figure 6 materials-13-00930-f006:**
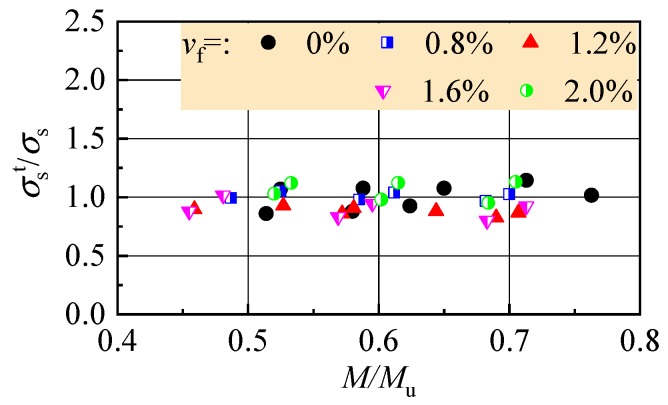
Ratio of tested to calculated values of the rebar stress at normal service.

**Figure 7 materials-13-00930-f007:**
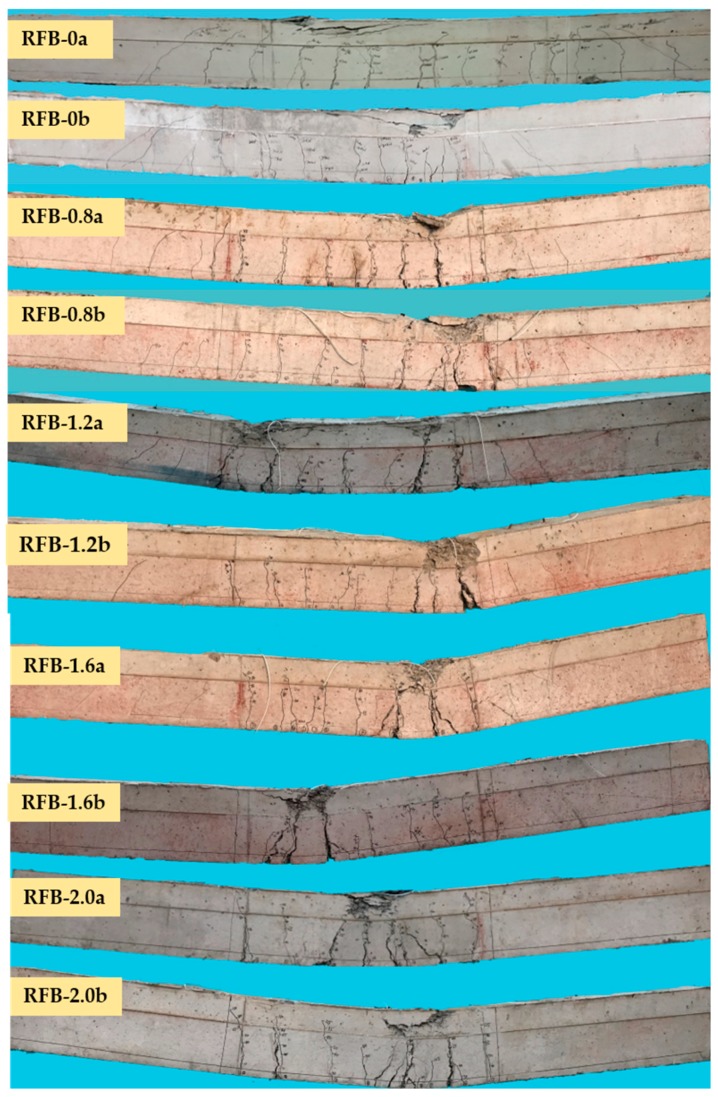
Crack distribution and failure patterns of the test beams.

**Figure 8 materials-13-00930-f008:**
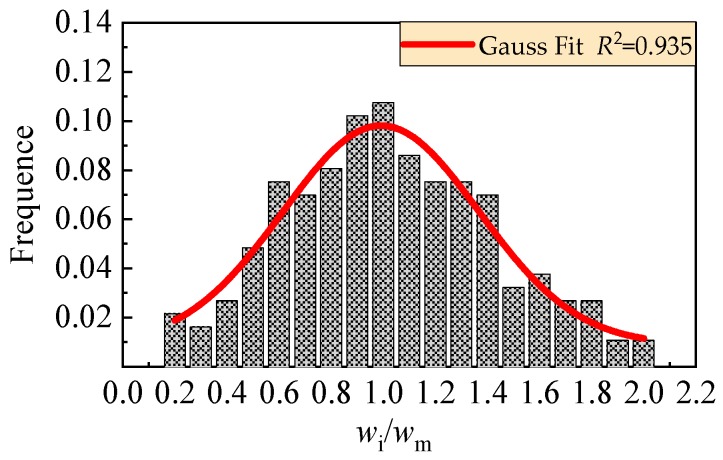
Statistical analysis of distribution of crack width.

**Figure 9 materials-13-00930-f009:**
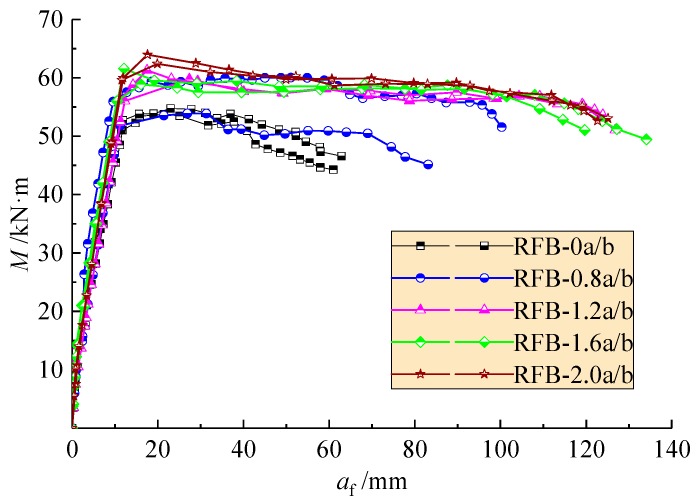
Mid-span deflection changed with moment of the test beams.

**Figure 10 materials-13-00930-f010:**
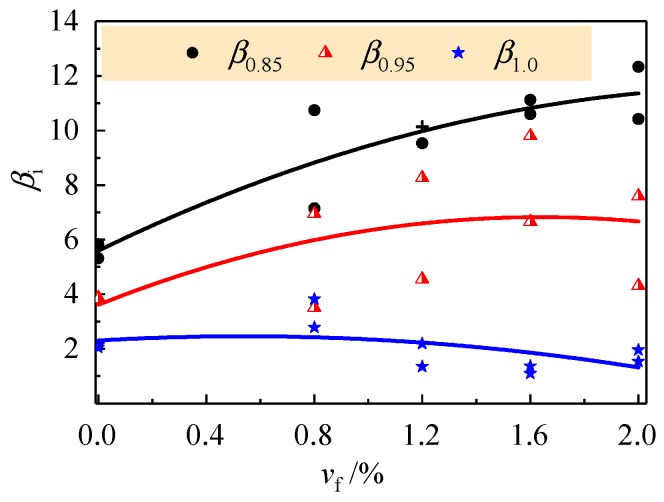
Flexural ductility indexes of the test beams with a varying *v*_f_.

**Figure 11 materials-13-00930-f011:**
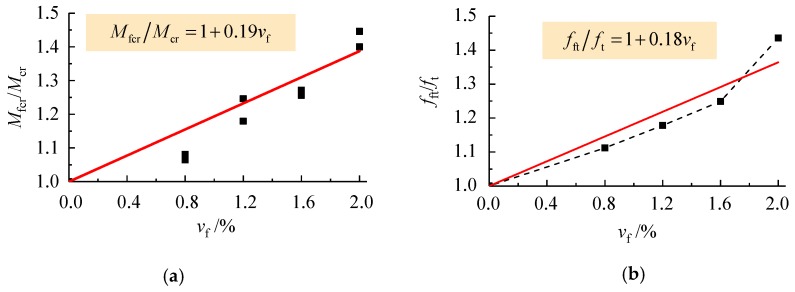
Comparison of the changes with *v*_f_: (**a**) *M*_fcr_; (**b**) *f*_ft_.

**Figure 12 materials-13-00930-f012:**
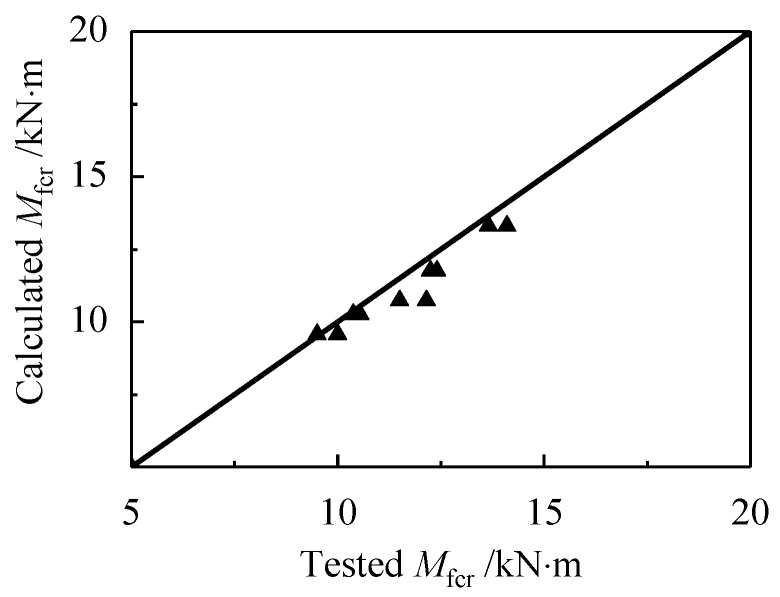
Test and calculated cracking moment of reinforced SFR-CRAC beams.

**Figure 13 materials-13-00930-f013:**
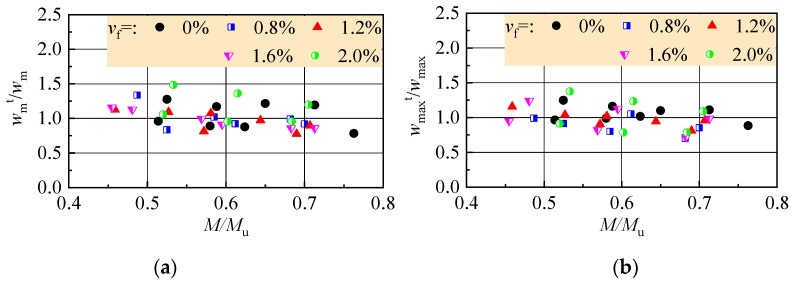
Ratio of tested to calculated values: (**a**) average crack width; (**b**) maximum crack width.

**Figure 14 materials-13-00930-f014:**
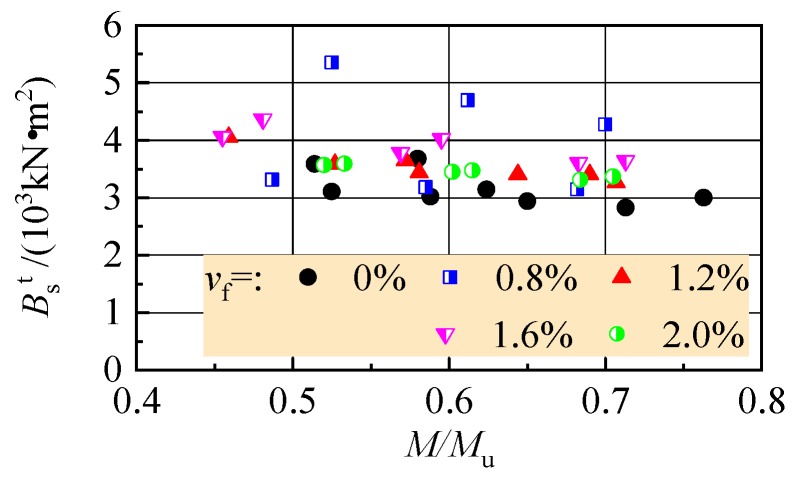
Tested flexural stiffness changed with moment of the test beams.

**Figure 15 materials-13-00930-f015:**
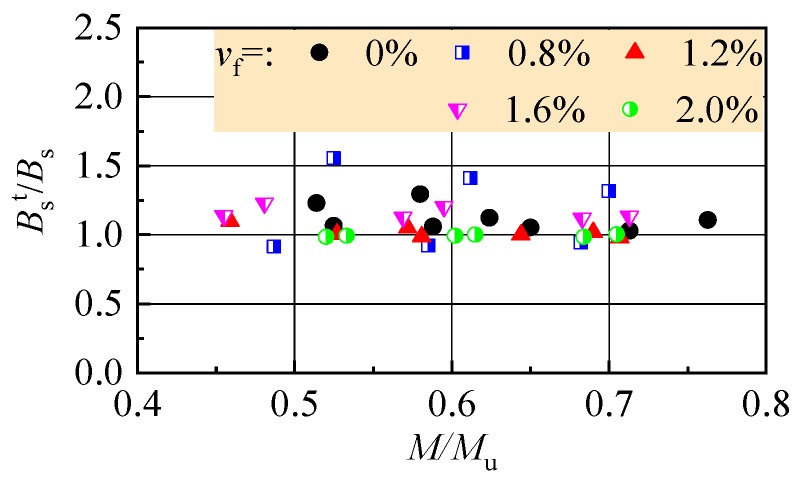
Test to calculation ratios of flexural stiffness.

**Figure 16 materials-13-00930-f016:**
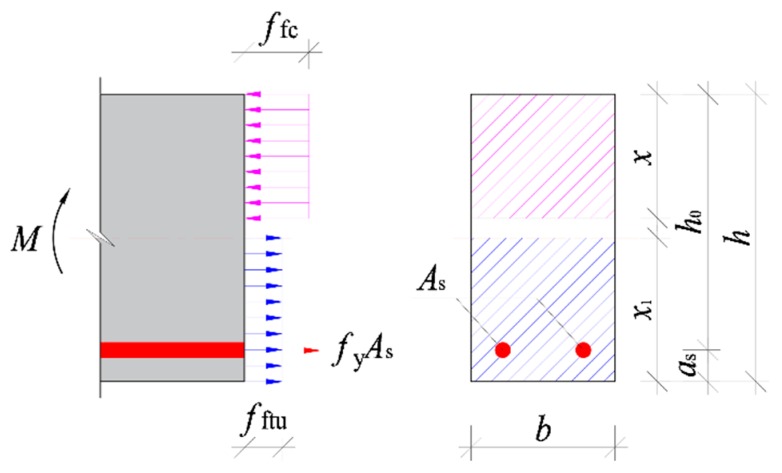
Numerical diagram of the ultimate moment at the cross-section.

**Figure 17 materials-13-00930-f017:**
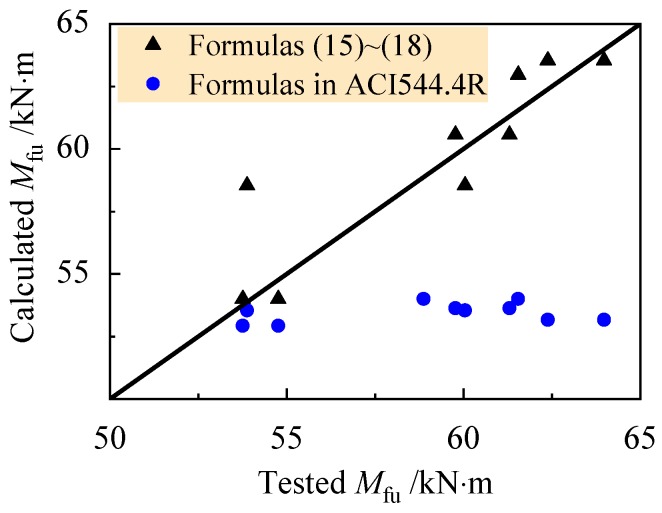
Tested and calculated ultimate moment of reinforced SFR-CRAC beams.

**Table 1 materials-13-00930-t001:** Basic properties of recycled and natural aggregates.

Identifier	Particle Size (mm)	Bulk Density (kg/m^3^)	Closed Compact Density (kg/m^3^)	Particle Density (kg/m^3^)	Water Absorption of 24 h (%)	Crushed Index (%)
Coarse RA	5–16	1293.3	1445.5	2673.8	5.10	14.30
Fine RA	0–5	1330	1470	2395.73	9.45	—
NA	16–25	1417	1592	2721.9	0.47	12.80

**Table 2 materials-13-00930-t002:** Mix proportion and basic mechanical properties of steel fiber reinforced composite-recycled aggregate concrete (SFR-CRAC) for test beams.

Identifier	*v*_f_ (%)	Dosage of Aggregates (kg/m^3^)			Additive (%)	Additional Water (kg/m^3^)	Basic Mechanical Properties (MPa)			
		Fine RA	Coarse FA	NA			*f* _cu_	*f* _fc_	*f* _ft_	*E* _c_
RFB-0a/b	0	709.1	587.5	391.7	0.3	41.8	36.2	27.7	2.41	2.08 × 10^4^
RFB-0.8a/b	0.8	725.6	563.5	375.7	0.3	41.8	39.3	30.2	2.68	2.61 × 10^4^
RFB-1.2a/b	1.2	733.8	551.5	367.7	0.4	41.8	39.9	30.6	2.84	2.85 × 10^4^
RFB-1.6a/b	1.6	742.0	539.5	359.6	0.5	41.8	36.8	32.4	3.01	2.29 × 10^4^
RFB-2.0a/b	2.0	750.3	527.5	351.6	0.6	41.8	38.4	28.5	3.46	2.53 × 10^4^

**Table 3 materials-13-00930-t003:** Main tested data of the cracking moment, ultimate moment, and the normal serviceable indices.

Identifier	*v*_f_ (%)	*M*_cr_ (kN·m)	*M*_u_ (kN·m)	*M*/*M*_u_	*M* (kN·m)	*σ*_s_ (MPa)	*w*_m_ (mm)	*w_max_* (mm)	*a*_f_ (mm)
RFB-0a	0.0	10.00	53.76	0.525	28.21	322.56	0.16	0.26	5.57
				0.588	31.61	364.35	0.17	0.28	6.43
				0.650	34.93	402.57	0.20	0.30	7.29
				0.713	38.36	469.35	0.22	0.34	8.32
RFB-0b	0.0	9.50	54.76	0.514	28.14	259.56	0.12	0.20	4.81
				0.580	31.78	298.83	0.13	0.24	5.30
				0.624	34.16	338.76	0.14	0.27	6.66
				0.763	41.79	454.44	0.16	0.30	8.55
RFB-0.8a	0.8	10.54	60.04	0.525	31.50	280.83	0.09	0.16	3.61
				0.612	36.75	336.76	0.13	0.24	4.80
				0.700	42.00	389.53	0.16	0.24	6.03
RFB-0.8b	0.8	10.38	53.88	0.487	26.25	209.37	0.10	0.12	4.86
				0.585	31.50	261.87	0.11	0.14	6.07
				0.682	36.75	312.48	0.14	0.16	7.17
RFB-1.2a	1.2	11.50	59.77	0.527	31.50	243.96	0.11	0.17	5.39
				0.581	34.75	270.56	0.13	0.20	6.20
				0.644	38.50	297.90	0.14	0.22	6.94
				0.707	42.25	328.71	0.15	0.26	7.93
RFB-1.2b	1.2	12.15	61.30	0.459	28.14	203.96	0.09	0.15	4.26
				0.572	35.05	258.94	0.10	0.18	5.89
				0.690	42.29	313.29	0.13	0.22	7.62
RFB-1.6a	1.6	12.40	58.87	0.481	28.31	232.52	0.09	0.16	3.98
				0.595	35.03	285.01	0.11	0.22	5.34
				0.713	42.00	346.67	0.14	0.26	7.07
RFB-1.6b	1.6	12.25	61.55	0.455	28.00	198.02	0.09	0.12	4.23
				0.569	35.00	250.16	0.12	0.16	5.68
				0.683	42.01	300.30	0.14	0.19	7.15
RFB-2.0a	2.0	13.65	63.98	0.520	33.30	279.31	0.10	0.14	5.73
				0.602	38.54	321.08	0.12	0.16	6.86
				0.684	43.80	364.71	0.15	0.20	8.11
RFB-2.0b	2.0	14.10	62.39	0.533	33.25	303.60	0.14	0.21	5.68
				0.615	38.37	365.40	0.17	0.25	6.78
				0.705	44.01	436.39	0.19	0.28	8.02

**Table 4 materials-13-00930-t004:** Test results of crack spacing compared with the calculation at normal serviceability.

Identifier		RFB-0a/b	RFB-0.8a/b	RFB-1.2a,b	RFB-1.6a,b	RFB-2.0a,b
Maximum (mm)		167, 161	161, 173	182, 164	142, 170	155, 160
Minimum (mm)		94, 123	104, 85	93, 71	88. 88	85, 85
Average (mm)	tested	118, 128	132, 110	111, 112	115, 110	110, 115
	Calculated	119	115	113	111	109
Tested/Calculated		0.990, 1.074	1.144, 0.954	0.978, 0.987	1.031, 0.986	1.003, 1.049

**Table 5 materials-13-00930-t005:** Calculation results of flexural ductility index of the test beams.

Identifier	*δ*_y_ (mm)	*δ*_0.85_ (mm)	*δ*_0.95_ (mm)	*δ*_1.0_ (mm)	*β* _0.85_	*β* _0.95_	*β* _1.0_
RFB-0a	10.46	55.50	40.56	21.40	5.31	3.88	2.05
RFB-0b	10.77	62.90	41.40	23.00	5.84	3.84	2.13
RFB-0.8a	9.34	100.30	64.90	35.70	10.74	6.95	3.82
RFB-0.8b	11.25	80.45	39.50	31.30	7.15	3.51	2.78
RFB-1.2a	12.50	126.80	103.35	27.35	10.14	8.27	2.19
RFB-1.2b	13.00	123.90	59.30	17.50	9.53	4.56	1.35
RFB-1.6a	11.73	130.50	115.00	16.05	11.12	9.80	1.37
RFB-1.6b	11.00	116.60	73.30	12.08	10.60	6.66	1.10
RFB-2.0a	11.46	119.46	49.35	17.60	10.42	4.31	1.53
RFB-2.0b	10.15	125.10	77.10	19.87	12.33	7.60	1.96

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
