# Peer review of "Bending Performance of Steel Fiber Reinforced Concrete Beams Based on Composite-Recycled Aggregate and Matched with 500 MPa Rebars"

_materials, 2020, doi:10.3390/ma13040930_

Round 1

Reviewer 1 Report

The article covers the topic of the Bending Performance of SFRC Beams Based on Composite-Recycled Aggregate and Matched with 500MPa Rebars.
The subject and the supporting experiments are informative and present added value to the body of knowledge on the subject area.
The topic of the article is in scope of journal. However, the following modification should be considered:

1. I suggest that abbreviation in the title 'SFRC' should be change to full name.
2. I suggest to add point 2 - Research significance - Please descibe here the main essence of the experiment. Please use the part of sentences from lines 77-87.
3. I sugget that literature survey should contain more thorough content. Consider to add some literature involved with this scientific area where authors define the major impact of the aggregate shape
on the mechanical properties and behavior of concrete; such as: https://doi.org/10.3390/ma11081372; https://doi.org/10.1016/j.conbuildmat.2019.117794
4. I think that the Young modulus of steel is higher than suggested in line 98. 2.05x10^4MPa is equal to 20.5 GPa. The Young modulus of steel reach this value from 205 to 210 GPa.
5. Table 1 - Please add abbreviation to all identifiers ( column 1) which you using then in Table 2. It must be clear that Recycled coarse aggregate is the same as 'RA'.
6. What type of cement was used in this tests? CEM I 42.5 R/NA/N? Please add more details with the content of cement.
7. Why fine aggregate has diameter from 0 to 5 mm? In general, fine aggregate is considered in class from 0 to 4 mm.Why are you trying to call fraction 4-5 mm as fine aggregate?
8. I suggest to add particle size distribution for coarse aggregate and fine aggregate.

9. Further analysis has been presented in proper way.

Author Response

Dear Professor,

Thanks very much for your comments.

The responses are made as follow, please check them.

I suggest that abbreviation in the title 'SFRC' should be change to full name.
Response: Ok. I suggest to add point 2 - Research significance - Please descibe here the main essence of the experiment. Please use the part of sentences from lines 77-87.
Response: Ok. I sugget that literature survey should contain more thorough content. Consider to add some literature involved with this scientific area where authors define the major impact of the aggregate shape on the mechanical properties and behavior of concrete; such as: https://doi.org/10.3390/ma11081372; https://doi.org/10.1016/j.conbuildmat.2019.117794
Response: Ok. The content is revised, and the literatures are added as references. I think that the Young modulus of steel is higher than suggested in line 98. 2.05x10^4MPa is equal to 20.5 GPa. The Young modulus of steel reach this value from 205 to 210 GPa.
Response: Thanks. It was a mistake. Indeed it is 205GPa. Table 1 - Please add abbreviation to all identifiers ( column 1) which you using then in Table 2. It must be clear that Recycled coarse aggregate is the same as 'RA'.
Response: Ok. It is done as your comment. What type of cement was used in this tests? CEM I 42.5 R/NA/N? Please add more details with the content of cement.
Response: As per China code GB175-2007 (common Portland cement), the cement is not complete equivalent to CEM42.5. The contents of cement has exhibited in referenced paper 15 published in Materials 2019. The referenced number of this paper is added at the end of the sentence. Why fine aggregate has diameter from 0 to 5 mm? In general, fine aggregate is considered in class from 0 to 4 mm. Why are you trying to call fraction 4-5 mm as fine aggregate?
Response: As per China code JGJ52, the particle size of fine aggregate ranges from 0 to 5mm, of which the weight of screen residue of 5mm particles is less than 10%. This is added in the revised manuscript. I suggest to add particle size distribution for coarse aggregate and fine aggregate.

Response: As the particle size distribution for coarse aggregate and fine aggregate has been exhibited in my published paper as list of references 15-17, and the papers 15 and 16 are published in Materials. To avoid repeat publishing of the same content, this content is not added.

Thanks again.

Best regards,

Fenglan Li

Reviewer 2 Report

The manuscript presents results from an investigation of the flexural performance of steel fiber reinforced composite-recycled aggregate concrete beams using a four-point load test. While the topic is interesting and the model has some merit, the overall manuscript does need some additional work to rise to the level of publication in the materials in the opinion of this reviewer. The following form the basis for this opinion.

General Comments:

---------------------

The introduction lists some relevant research but fails to present a scientific review. Please consider to rewrite and clarify the motivation, objectives, and significance of this study. The introduction does not identify the knowledge gap that the authors are trying to address with their research. The objective statements are rather vague and lack projected outcomes or how the paper will assist practitioners. Some unnecessary descriptions of the test results should be removed, and more in-depth analysis should be added. The quality of the figures must be improved. Your conclusions are not well supported by your experimental data and analysis. Please clearly list the new, key findings supported by the experimental investigation. It reads like a report overall. The presentation is poor, and there lack of in-depth analysis and discussions. The manuscript needs a pretty thorough grammatical/spelling review. There are numerous sentences that are either incorrect grammatically, do not make sense, to have misspelled. Some sentences need to be revised. References’ format needs to be checked. More experimental results are needed to verify the accuracy of the modified equations proposed by the authors.

Author Response

Dear Professor,

Thanks very much for your comments.

The responses are made as follow, please check them.

The introduction lists some relevant research but fails to present a scientific review. Please consider to rewrite and clarify the motivation, objectives, and significance of this study. The introduction does not identify the knowledge gap that the authors are trying to address with their research. The objective statements are rather vague and lack projected outcomes or how the paper will assist practitioners.

Response: Ok. The introduction is revised as your comment. Please check the RED parts of the revised manuscript.

Some unnecessary descriptions of the test results should be removed, and more in-depth analysis should be added. The quality of the figures must be improved.

Response: Ok. Figs. 1,2 and 6 are renewed. Please check the RED parts of the revised manuscript.

Your conclusions are not well supported by your experimental data and analysis. Please clearly list the new, key findings supported by the experimental investigation. It reads like a report overall.

Response: Ok. It is revised as your comment. Please check the revised manuscript.

The presentation is poor, and there lack of in-depth analysis and discussions.

Response: Ok, the presentation and discussion are revised. Please check the RED parts of the revised manuscript.

The manuscript needs a pretty thorough grammatical/spelling review. There are numerous sentences that are either incorrect grammatically, do not make sense, to have misspelled. Some sentences need to be revised. References’ format needs to be checked.

Response: Ok. We do our best to review and revise the manuscript. Please check.

More experimental results are needed to verify the accuracy of the modified equations proposed by the authors.

Response: Yes. With the deep investigation of SFR-CRAC, we will provide more experimental results to verify the accuracy of the modified equations. At present, no more studies performed on reinforced SFR-CRAC beams.

Best regards,

Fenglan Li

Reviewer 3 Report

The authors presented an interesting issue related to the study of the flexural performance of steel fiber reinforced composite-recycled aggregate concrete  beams matched with 500MPa longitudinal rebars. However the following comments should be solved before trying to publish the paper.

line 16. Double dot. Line 57 - It would be advisable to give examples of recipes (e.g. aggregate percentage) and example properties (numerical values) for the cited literature. The abbreviations SFR-CRAC and CRAC should be explained in the "Experimetal work" section, even though they have already been explained in indroduction. Subchapter 2.2. The manufacturer's name, city and country of each material and device should be provided. Table 1. "Crushed index (%)" - for the given numerical value, remove "%" (First line). Line 121 - 42% is the percentage (by volume) of Fine RA in relation to the whole mix? Subchapter 2.3. After what time the tests were performed? How long did the samples mature? Fig. 2 - There are two schemes with different strain gauges spacing. Please mark these schemes as "a" and "b" and sign what they relate to. Tabble. 3. it would be advisable to describe the symbols used in the table. Table 4. Instead of commas, there should be a dot as a separator. Table 5. "Identifier" should be written on one line.

Author Response

Dear Professor,

Thanks very much for your comments.

The responses are made as follow, please check them.

(1) line 16. Double dot. Line 57 - It would be advisable to give examples of recipes (e.g. aggregate percentage) and example properties (numerical values) for the cited literature.

Response: Ok. Please check the RED parts of the revised manuscript.

(2) The abbreviations SFR-CRAC and CRAC should be explained in the "Experimental work" section, even though they have already been explained in introduction.

Response: Ok. It is added.

(3) Subchapter 2.2. The manufacturer's name, city and country of each material and device should be provided.

Response: Ok. They are provided.

(4) Table 1. "Crushed index (%)" - for the given numerical value, remove "%" (First line).

Response: Ok. It is deleted.

(5) Line 121 - 42% is the percentage (by volume) of Fine RA in relation to the whole mix?

Response: No, it is the percentage in mass of total aggregates. This revised in the manuscript.

(6) Subchapter 2.3. After what time the tests were performed? How long did the samples mature? 

Response: Tests were conducted after the beams cured for 28 days. This information is added in the manuscript.

(7) Fig. 2 - There are two schemes with different strain gauges spacing. Please mark these schemes as "a" and "b" and sign what they relate to.

Response: Ok. It is signed and added the explanation in text.

(8) Table. 3. it would be advisable to describe the symbols used in the table.

Response: The description of the symbols presented in Table 3 are in the first paragraph of Section 4.3 in the revised manuscript.

(9) Table 4. Instead of commas, there should be a dot as a separator.

Response: In Table 4, the commas were used to separate the data of two test beams marked as a and b. By using dot may lead a mistake as one data.

(10) Table 5. "Identifier" should be written on one line.

Response: Ok.

Best regards,

Fenglan Li

Round 2

Reviewer 1 Report

All remarks have been considered by authors. Errors have been eliminated. The authors responded to all comments of the reviewer.
The current version is satisfactory for reviewer.

In lines 155/156 please do not use the abbreviation 'd', I suggest to change it to 'days'.

In my opinion, article could be published.

Reviewer 2 Report

 - All the review comments have been successfully addressed and acceptance of the paper in its current form is recommended.